# Acute Toxicity and Anti-Inflammatory Activity of *Trattinnickia rhoifolia* Willd (Sucuruba) Using the *Zebrafish* Model

**DOI:** 10.3390/molecules27227741

**Published:** 2022-11-10

**Authors:** Agerdânio Andrade de Souza, Brenda Lorena Sánchez Ortíz, Swanny Ferreira Borges, Andria Vanessa Pena Pinto, Ryan da Silva Ramos, Igor Colares Pena, Rosemary de Carvalho Rocha Koga, Carla Estefani Batista, Gisele Custódio de Souza, Adriana Maciel Ferreira, Sergio Duvoisin Junior, José Carlos Tavares Carvalho

**Affiliations:** 1Post-Graduate Program in Pharmaceutical Innovation, Pharmacy Course, Department of Biological and Health Sciences, Federal University of Amapá, Rodovia Juscelino Kubitschek, Macapá CEP 68903-419, Amapá, Brazil; 2Indigenous Intercultural Licensing Course, Binational Campus, Federal University of Amapá, Rodovia BR 156, No. 3051, Universidade, Oiapoque CEP 68980-000, Amapá, Brazil; 3Research Laboratory of Drugs, Department of Biological and Health Sciences, Federal University of Amapá, Rodovia Juscelino Kubitschek, km 02, Macapá CEP 68903-419, Amapá, Brazil; 4Graduate Program in Biotechnology and Biodiversity-Network BIONORTE, Federal University of Amapá, Macapá CEP 68903-419, Amapá, Brazil; 5Laboratory of Modeling and Computational Chemistry, Department of Biological and Health Sciences, Federal University of Amapá, Macapá CEP 68902-280, Amapá, Brazil; 6School of Technology, University of the State of Amazonas–UEA, Manaus CEP 69050-020, Amazonas, Brazil; 7University Hospital of the Federal University of Amapá, R. do Estádio Zerão, Macapá CEP 68902-336, Amapá, Brazil

**Keywords:** ethnomedicine, *Trattinnickia*, anti-inflammatory, preclinical trials, molecular docking

## Abstract

The species *Trattinnickia rhoifolia* Willd, (*T. rhoifolia*), which belongs to the *Burseraceae* family, is widely used in ethnopharmacological cultural practices by traditional Amazonian people for anti-inflammatory purposes, sometimes as their only therapeutic resource. Although it is used in teas, infusions, macerations and in food, the species is still unexplored in regard to its pharmacophoric potential and chemical profile. Therefore, the aim of this study was to conduct a phytochemical characterization of the hydroethanolic extract of *T. rhoifolia* leaves (HELTr) and to evaluate the acute toxicity and anti-inflammatory activity of this species using zebrafish (*Danio rerio*). The extract was analyzed by gas chromatography–mass spectrometry (GC-MS). The evaluation of the acute toxicity of the HELTr in adult zebrafish was determined using the limit test (2000 mg/kg), with behavioral and histopathological evaluations, in addition to the analysis of the anti-inflammatory potential of HELTr in carrageenan-induced abdominal edema, followed by the use of the computational method of molecular docking. The phytochemical profile of the species is chemically diverse, suggesting the presence of the fatty acids, ester, alcohol and benzoic acid classes, including propanoic acid, ethyl ester and hexadecanoic acid. In the studies of zebrafish performed according to the index of histopathological changes (IHC), the HELTr did not demonstrate toxicity in the behavioral and histopathological assessments, since the vital organs remained unchanged. Carrageenan-induced abdominal edema was significantly reduced at all HELTr doses (100, 200 and 500 mg/kg) in relation to the negative control, dimethyl sulfoxide (DMSO), while the 200 mg/kg dose showed significant anti-inflammatory activity in relation to the positive control (indomethacin). With these activities being confirmed by molecular docking studies, they showed a good profile for the inhibition of the enzyme Cyclooxygenase-2 (COX-2), as the interactions established at the sites of the receptors used in the docking study were similar to the controls (RCX, IMN and CEL). Therefore, the HELTr has an acceptable degree of safety for acute toxicity, defined in the analysis of behavioral changes, mortality and histopathology, with a significant anti-inflammatory action in zebrafish at all doses, which demonstrates the high pharmacophoric potential of the species. These results may direct future applications and drug development but still require further elucidation.

## 1. Introduction

Medicinal plants are known for their therapeutic properties, since they are used as a natural method of treating and curing diseases [1]. Many communities and ethnic groups around the world have become disseminators and users of herbs and medicinal plants, especially among the traditional peoples and communities (TPCs) of the Amazon, for whom the forest is sometimes the only therapeutic resource due to regional isolation [2,3]. Therefore, Amazonian biodiversity represents a source for the discovery of new anti-inflammatory drugs with high levels of efficiency, reduced side effects and high efficacy due to the variety of existing sources [4,5].

In this way, the intrinsic nexus between the forest and the TPCs (that is, between plant species and the cultural practices of ethnomedicine and ethnopharmacology) has become interesting to the pharmaceutical industries, especially those species whose pharmacological potential is still unexplored [4].

Therefore, the intrinsic nexus between the forest and TPCs (that is, between plant species and the cultural practices of ethnomedicine and ethnopharmacology) has been an object of interest of pharmaceutical industries, with emphasis on those species whose pharmacological potential is still unexplored, representing a source for new drugs that are more effective, less expensive and safer, with varied production mechanisms. Anti-inflammatory drugs are divided into two categories, steroidal and non-steroidal anti-inflammatory drugs (NSAIDs), and those available on the market today include diclofenac, ibuprofen and several others, which have undesirable side effects. Even so, these drugs are not accessible to Amazonian populations, who, therefore, resort to native plant species as a source of bioactive agents with anti-inflammatory actions [4,5].

Among the latter category, the species *T. rhoifolia*, which belongs to the *Burseraceae* family, is noteworthy for being a long woody tree and for exuding a very aromatic oleoresin [4,5,6]. Its peculiarities contribute to the creation of different popular names for the species in the Amazon region, such as: “*breu sucuruba*”, “*almescla*”, “*gugul*” and “*maaliol*”, among others [4,7]. Furthermore, it is a native plant that has a phytogeographic occurrence in the Amazon, which is not endemic in Brazil and presents opportunities for extensive medicinal manipulation and ethnopharmacological use among the TPCs of the Amazon [4,6,7,8].

Although *T. rhoifolia* is widely used in folk medicine in the Brazilian Amazon as an analgesic, anti-inflammatory, expectorant and healing agent as part of ethnomedicinal practices, scientific studies on the toxicity, phytochemical characterization and pharmacological potential of this species are rare [4,9]. Among the studies that evaluate its pharmacophoric potential, research on immunomodulation stands out [10,11]. These studies demonstrated the expressive antioxidant, antifungal and anti-inflammatory actions of the compounds in the leaves and the resin of the plant [4,11,12,13], which are rich in monoterpenes, alkaloids, terpenoids, sesquiterpenes, ketones, flavonoids, bioflavonoids, oxyphytosterols and acids, the latter being less frequent [4,7,8].

Other studies on amentoflavone isolated from leaves revealed the anti-HIV properties of *T. rhoifolia*, which characterizes the species as one of the most promising for the development of new drugs, in regard to its phytochemical profile [10,11,14,15]. Nonetheless, the chemical mapping of the constituent secondary metabolites and knowledge of the toxicity and the anti-inflammatory potential of the species still represent scientifically unexplored fields. However, the population of the Amazon region uses the species as a means of treatment for different diseases and as an anti-inflammatory [16,17,18].

Additionally, there are difficulties involved in biomonitoring the species *T. rhoifolia* as a result of the lack of research on pharmaceutical innovation in the Brazilian Amazon. These circumstances impair the study of its potential [4,19]. In this sense, the isolation of the region does not only affect TPCs, but it is also reflected in research centers, given that the distance of the region from large centers makes inputs more expensive and poses difficulties for the conduction and maintenance of in vivo tests [17,20].

Hence, in order to minimize the cost and optimize research with in vivo tests, robust and innovative experimental models have been used, such as zebrafish, which has numerous advantages when compared to the classic pre-clinical in vivo models, such as rodents. Among the advantages, firstly, the short life cycle and high fecundity rate (production potential: up to 100 eggs/day) stand out, benefiting its production and, subsequently, its use in biological assays over a short period of time [18,21,22,23]. Moreover, the small size, associated with the low body weight of the adult zebrafish, enables the use of smaller quantities of the tested compounds, since the calculation is based on the animal’s body weight, being considered one of the most relevant advantages in regard to its use, whether for the screening of new compounds, toxicity tests, the screening of new anti-inflammatory substances, or other types of natural product research [23,24,25]. This is because inflammation is an innate biological reaction of the organism, configured by an intricate network of biochemical and cellular events in response to damage, injury, infections, immunological reactions and harmful agents [23,26,27].

Among the studies that evaluate the anti-inflammatory activity in zebrafish, the one examining nanoemulsions through the use of *Rosmarinus officinalis* (*R. officinalis*) essential oils stands out [24,28], in which the ability to potentiate the anti-inflammatory action was demonstrated. The said activity had already been observed at certain doses of fatty-amides (N-alkylamides) [25,29], which were effective in reducing carrageenan-induced abdominal edema in zebrafish [30]. Due to this finding, studies of native Amazonian plant species and their anti-inflammatory potential in the context of acute inflammation may result in the discovery of compounds of various chemical classes with an alternative or desirable mechanism of action in relation to the therapeutic profile examined in the study [4,28,30].

In this regard, in the Amazonian context, computational methods, such as molecular docking, which are known for their speed and low cost, are used as an efficient strategy in the process of the discovery of new drug candidate molecules and in the elucidation of their mechanisms of action. The use of this tool makes it possible to identify the ligands with the highest affinity for a given molecular target [31,32].

Thus, the present study focused on the chemical characterization of *T. rhoifolia* leaves, as well as the evaluation of its acute toxicity and anti-inflammatory effect, in an experimental model of adult zebrafish, combined with the computational method of molecular docking to further elucidate the possible mechanisms of the species’ anti-inflammatory actions already reported by TPCs. Thus, this scientific research will confirm the reports and cultural practices of ethnomedicine and ethnopharmacology in the Amazon region regarding this species and its technical-scientific validation through experimentation.

## 2. Results

The HELTr was analyzed by GC-MS in order to allow us to better understand the chemical profile of secondary metabolites present in this species. These elements were identified by the total ion chromatogram (TIC) (Figure 1), considering the similarities of more than 70% of the experimental spectra in relation to those of the National Institute of Standards and Technology Mass Spectrometry (NIST MS) Library, which are demonstrated in Figure 1.

The GC-MS of the HELTr shows chemically diverse metabolites, suggesting the presence of fatty acids, esters, alcohols and benzoic acid with a similarity higher than 70%, totaling 12 compounds (Table 1). Of the total value, there are 7 compounds with over 90% similarity. When listing the compounds indicated in the profiles of the GC-MS analyses performed on the HELTr, the presence of propanoic acid, ethyl ester (Figure 2a); hexadecanoic acid, ethyl ester (Figure 2b); 1-propanol 2-methyl (Figure 2c); and *n*-propyl acetate (Figure 2d) can be observed. In addition, the chemical compounds propane, 2,2-diethoxy (Figure 2e); *n*-hexadecanoic acid (Figure 2f); benzophenone (Figure 2g); and propane,1,1,3-triethoxy (Figure 2h) were verified. Moreover, acetic acid, ethenyl ester (Figure 2i) [20]; 1-butanol, 3-methyl (Figure 2j); 3-hydroxybenzoic acid (Figure 2k); and decanoic acid, ethyl ester (Figure 2l) were likewise indicated via the chromatogram and NIST MS Library [33,34,35,36,37,38,39,40,41,42,43,44,45,46].

### 2.1. Study of Acute Toxicity

#### 2.1.1. Behavioral Analysis

Oral treatment with a 2000 mg/kg dose of the HELTr triggered a few behavioral changes in the zebrafish, as shown in Table 2. The percentage of changes was higher for animals treated with the HELTr dose than animals treated with the control (2 µL of DMSO). The stress signs recorded were increased swimming activity, the animal resting on the bottom of the aquarium and loss of posture. No deaths were recorded within 48 h.

#### 2.1.2. Histopathological Analysis

Treatment with the 2000 mg/kg dose of the HELTr in the control group caused a few histopathological changes in the zebrafish liver, intestine and kidneys. Regarding the IHC, where values ranging from 0–10 are considered normal, the analyses performed for all the organs studied yielded values less than 10 (Figure 3), which characterize these organs as normal, as the alterations recorded did not alter their normal functioning.

Only Stage I alterations were registered in all organs. In the liver, cytoplasmic vacuolization and nuclear atypia were observed (Figure 4B). In the intestine, we observed dilation of the vessels present in the villi and the infiltration of leukocytes (Figure 4D). In the kidneys, mild tubular hyaline degeneration and tubular cell hyperplasia, signs that do not compromise the normal functioning of these organs, were observed (Figure 4F).

### 2.2. Effect of the HELTr on Carrageenan-Induced Abdominal Edema

The HELTr presented an inhibitory effect on carrageenan-induced abdominal edema (Figure 5, Figure 6 and Figure 7). For all HELTr doses (100 mg/kg, 200 mg/kg and 500 mg/kg), including the positive control (indomethacin 10 mg/kg), this effect started during the 1st h and remained throughout the entire experimental period (0–5 h) (Figure 5).

At the maximum peak of edema (5th hour) (Figure 7), the reduction corresponded to 88.1%, 97.6% and 92.0% for the doses of 100 mg/kg, 200 mg/kg and 500 mg/kg, respectively. Indomethacin 10 mg/kg (positive control) showed an inhibition of 75.7% at the maximum peak of edema.

It was observed that all the treatments with the HELTr significantly reduced abdominal edema to similar extents, with no statistically significant difference among the doses tested. On the other hand, all HELTr doses showed a statistically significant difference in relation to the negative control group (DMSO+CAR) (*p* ≤ 0.05) at all the times evaluated.

Hence, regarding the positive control group (indomethacin 10 mg/kg), all HELTr doses showed no statistical difference. Only the dose of 200 mg/kg showed a statistically significant difference (*p* ≤ 0.05) at the maximum peak of edema (5th hour) (Figure 6).

### 2.3. Histopathological Analysis of the Inflammation

#### 2.3.1. Liver

In the liver tissue, the PBS (intraperitoneal, i.p.) and saline (p.o.) group showed only level I changes, such as cytoplasmic vacuolization (Figure 8). The IHC was 0.50 ± 0.57, which indicates the normality of the organ after PBS application.

The animal groups that received carrageenan (i.p.) and indomethacin 10 mg/kg (p.o.), as well as HELTr doses of 100, 200 and 500 mg/kg (p.o.), also demonstrated a low IHC (0.75 ± 0.50, 0.25 ± 0.50, 1.25 ± 0.50 and 2.0 ± 0.81, respectively). The main histopathological alterations observed were cytoplasmic vacuolization and nuclear atypia.

In the animals treated with carrageenan (i.p.) and 2 µL of DMSO (p.o.), the liver IHC was quite high (31.75 ± 4.54), which classified the alterations in this organ as ranging from moderate to severe (levels I and II). The main level I alterations observed were cytoplasmic vacuolization and loss or atypia of the cellular and nuclear contour. At level II, hyperemia, cytoplasmic degeneration and cell degeneration were observed (Figure 9).

#### 2.3.2. Intestine

No histopathological change was observed in the intestines of animals treated with PBS (i.p.) and saline (p.o.) (Figure 8). The animals that received carrageenan (i.p.) and indomethacin 10 mg/kg (p.o.), in addition to HELTr doses of 100, 200 and 500 mg/kg (p.o.), had an IHC higher than 10 (13.25 ± 1.70, 12.75 ± 2.06, 12.25 ± 0.95 and 11.75 ± 1.25, respectively), which classifies the alterations in this organ as ranging from mild to moderate. The main level I histopathological changes observed were the detachment of the epithelial lining from the apex of the intestinal villi, epithelial cell hypertrophy, vacuolization of the enterocytes and leukocyte infiltration. As regards level II changes, we observed the displacement of the intestinal lamina propria (Figure 9).

In animals treated with carrageenan (i.p.) and 2 µL of DMSO (p.o.), the intestine IHC was high (to 29.75 ± 6.70), which classified the alterations in this organ as ranging from moderate to severe. The main level I alterations observed were the detachment of the epithelial lining from the apex of the intestinal villi, epithelial cell hypertrophy, vacuolization of enterocytes and leukocyte infiltration. At level II, the displacement of the lamina propria, desquamation of the intestinal mucosa and villous degeneration were observed (Figure 7).

#### 2.3.3. Kidneys

In the kidneys, it was observed that the animal groups that received PBS (i.p.) and saline solution (p.o.) and those that received carrageenan (i.p) and 100, 200 and 500 mg/kg doses of the HELTr (p.o.) had the lowest IHC (0.50 ± 0.57, 0.75 ± 1.50, 1.75 ± 0.95 and 1.75 ± 1.70, respectively). These groups showed only level I alterations, such as mild tubular hyaline degeneration, a decrease in the Bowman’s capsule space area and dilated glomerular capillaries.

The highest IHC was observed in the groups that received carrageenan (i.p.), 2 µL of DMSO (p.o.) (31.75 ± 9.39, moderate to severe changes) and indomethacin 10 mg/kg (p.o.), (11.75 ± 2.06, mild to moderate changes). Level I and II changes were recorded. The main level I changes observed were mild tubular hyaline degeneration, a decrease in the Bowman’s capsule space area, dilated glomerular capillaries, tubular cell hypertrophy and an increase in the tubular lumen. As for level II, severe tubular hyaline degeneration, nuclear degeneration of the tubular cells and glomerular degeneration were observed (Figure 9).

### 2.4. Analysis for Molecular Docking

#### Selection of Enzyme and Inhibitor Structure

Validation tests of molecular docking protocols were performed by superimposing the crystallographic structures of the reference compounds onto the biological targets, so as to search for a bioactive conformation similar to the original crystallographic structure of the COX-2 ligands (PDB ID: 5KIR and 4COX) (Figure 10).

The validation results were considered satisfactory where the relative position of the crystallographic ligand and the coupled ligand were found to be similar (Figure 1). By retrieving the pose of each COX-2 inhibitor (RCX and IMN), it was possible to perform the validation of the molecular docking protocols by calculating the root mean square deviation (RMSD) values of 0.881 and 0.740 Å, respectively. Thus, the specific literature states that when the RMSD values are ≤2 Å, the docking protocol is considered satisfactory, as it shows similarity to the experimental model [47,48,49,50].

The molecular docking pose obtained enabled RCX to bind to amino acid residues in the COX-2 binding site compared to the crystallographic pose. In the binding site, the interactions take place around the α-helix between the amino acid residues Thr88-Tyr91 and Val344-Leu352 and in the β-sheet between the amino acid residues Leu353-Asp362 and Ala516-Gly519. In RCX, hydrogen bonds with residues Ile517 and Phe518 can be observed, as well as planar hydrophobic type interactions with Val349 and Leu352 [51].

Compared to the crystallographic pose, the molecular docking pose of IMN interacts with amino acid residues at the COX-2 binding site around the α-helix between amino acid residues Val116-Arg120, Tyr348-Leu352, Glu380-Leu384 and Met522-Ala527, and at the β-sheet between amino acid residues Tyr355-Leu359 and Tyr385-Trp387. In IMN, one can see common hydrogen bonds with residue Tyr355, an attractive charge with Arg120 and hydrophobic interactions with residues Val349, Leu352, Val523 and Ala527.

To evaluate whether the obtained conformations show a higher binding affinity than the specific ligand for each COX-2 therapeutic target (*Homo sapiens* and *Mus musculus* organisms), we coupled each therapeutic target on COX-2 with a specific control ligand and the potential molecules. The control ligands IMN and RCX showed the best binding affinity values for COX-2 than the other ligands evaluated in both organisms.

The control, CEL, presented a better binding affinity value in the *Homo sapiens* organism (5KIR) because it is a drug with a previously proven and commercialized anti-inflammatory activity; however, studies warn about toxicity effects (mutagenicity alert) and adverse effects when administered orally as ulcers, extrapyramidal and carcinogenic [52,53]. The potential ligand B (Figure 11) showed the best binding affinity value (−8.1 Kcal/mol), followed by I (−6.3 Kcal/mol), among the molecules evaluated in the case of COX-2 (*Homo sapiens* organism), having a range of ±1.9 Kcal/mol and ±0.8 Kcal/mol compared to RCX and IMN, respectively.

The best-evaluated inhibitors in respect to the binding affinity were B and I, in which the interactions were similar to the observed in RCX and IMN controls for the residue was Val116, and this interaction contribution may help in stabilization at the binding site [54].

The control molecules and potential inhibitors showed no violations, except for I and A, but these are in accordance with Lipinski’s rule of five, which is in line with the studies conducted previously [55]. Ligands with fewer or, preferably, no violations of these rules are more suitable for oral administration (Table 3).

On the other hand, in the case of COX-2, there are some interactions that create more accessibility in the lipophilic channel in this isoform than in COX-1, which can be observed, indicating a greater ease of interaction with the active site of COX-2 via Phe518.

Molecule B, with the free energy of −8.1 Kcal/mol, presents hydrophobic category interactions (pi-alkyl type) with Leu352 and Val523 and another hydrophobic (pi-sigma) interaction with Val523, the latter also being similar in the case of the RCX control. The second potential molecule I (−6.3 Kcal/mol) shows a hydrogen bridge with the amino acid residue Ser353 and hydrophobic (alkyl) type interactions with Val116, Ala527 and Leu531. The conformation of the inhibitors at the binding site is influenced by the distances of the interactions with the amino acid residues.

The ligand IMN showed the highest binding affinity (−10.1 Kcal/mol) at the COX-2 receptor (*Mus musculus* organism), followed by CEL (−9.9 Kcal/mol). The potential inhibitor A (Figure 12) showed binding affinity values close to the controls, for which the differences compared to IMN, RCX and CEL were ± 2.1, ± 1.9 and ± 0.3 Kcal/mol, respectively.

Molecule A, with the free energy of −8.0 Kcal/mol, presents hydrophobic category interactions (Alkyl type) with the amino acid residues Val349, Leu352, Tyr387, Phe381 and Trp387, the latter two being important hydrophobic interactions similar in the case of the IMN control. We can also observe a higher number of interactions in the case of indomethacin, which is a potent non-selective inhibitor of the COX enzyme, being a key element of the arachidonic acid cascade [54].

## 3. Discussion

Knowing that the species *T. rhoifolia* is widely used by TPCs in the Amazon region in ethnomedicinal practices as a powerful anti-inflammatory, in accordance with the traditional reports [4], this study obtained unprecedented results about the *species T. rhoifolia*, especially regarding the elucidation of the chemical profile through in vivo tests using non-target organisms—zebrafish. The experiments were based on reports of traditional use, aiming to fill unexplored gaps in scientific research on the anti-inflammatory action of the species.

The HELTr, based on similarity in GC-MS analysis, indicated the presence of 12 compounds (Table 1), which have already been characterized in other species of the *Burseraceae* family. Among the identified compounds, we highlight studies involving the hydroethanolic extracts of *Protium confusum* [34] and *Boswellia negligencia* [35] leaves, both species of the *Burseraceae* family, which identified the presence of hexadecanoic acid and ethyl ester (Figure 2A) via GC-MS, with similarities of ≥70%. In the HELTr, the corresponding standard was raised to ≥95%, as stated in the analyses of the NIST/MS Library. Additionally, in accordance with the literature, the compound identified in the studied extract has a high anti-inflammatory activity [56,57], and it is non-toxic, based on extensive tests using zebrafish embryos [57]. The HELTr analyses indicated a 98% similarity in the case of propanoic acid and ethyl (Figure 2B) in comparison with the literature in which the identifications of this chemical constituent in the Burseraceae family are described, in the case of both *Canarium album* L. (*C. album*) [58,59] and *Commiphora wightii* (“Guggul”) [37,38]. Additionally, previously performed chemical and pharmacological investigations of leaf extracts, young stems and gum resin showed anti-inflammatory [58], antioxidant and anti-aging actions [58], with a wide range of use in the functional preparation of nutraceuticals [58,59].

Other research involving qualitative and quantitative analyses using extracts from the leaves and resin of *T. rhoifolia* [4,21] and *C. album* [49], both from the *Burseraceae* family, confirmed the presence of *n*-hexadecanoic acid (Figure 2C), [45,46]. In the present study, the analyzed HELTr was 90.9% similar to the aforementioned compound.

In regard to this finding, it is noteworthy that previous studies carried out with extensive pharmacological investigations and experimental analyses in vivo and in vitro demonstrated that the *n*-hexadecanoic acid molecule has proven pharmacophoric properties, such as potent anti-inflammatory [59,60,61], anticancer [59,60], antibacterial [59], detoxifying [59], antioxidant [59,60,62] and hepatoprotective [59] actions, in addition to anti-hepatitis B [59,60] and anti-HIV [59,63] activities.

Additionally, the chemical profile of the HELTr indicated the presence of benzophenone (Figure 2D), with a similarity of 90%, the same compound already identified in *Bursera simaruba* leaves (*Burseraceae*) via GC-MS analysis [35,36]. Studies demonstrate the remarkable anticancer activity [64,65] of this compound, with a proven anti-inflammatory action, as demonstrated by the inflammation of paw edema induced by carrageenan [66].

The HELTr analyses also showed an 80% similarity in the case of the acetic acid ethenyl ester, reported to be responsible for several biological activities, such as anti-inflammatory, anti-obesity, antineoplastic and antidiabetic actions [20,67]. Other studies carried out on four species of the *Burseraceae* plant family in which the acetic acid ethenyl ester was characterized—*Boswellia dalzielli, Boswellia carteri* (olibanum gum), *Commiphora mukul* and *Commiphora incisa*—demonstrated its anti-inflammatory activities, since the acid was able to significantly reverse a carrageenan-induced inflammatory response in both maximal and rat paw edema [67,68].

3-Hydroxybenzoic acid, already characterized in *Boswellia ovalifoliolata* Bal & Henry (*Burseraceae*) by monitoring the isolated compound, was able to trigger antioxidant, anti-inflammatory, antiviral and anticancer activities [15,69].

In turn, the similarities of 1-propanol 2-methyl [39], *n*-propyl acetate [46,51], propane, 2,2-diethoxy [40], propane,1,1,3-triethoxy [41,42], 1-butanol, 3-methyl [43], decanoic acid and ethyl ester [14,46] were greater than 70% in the HELTr. According to the NIST/MS Library, similar results were found for other plant species. However, research is insufficient in relation to the pharmacological actions of these compounds, which, for the species studied here, were described for the first time.

Thus, all compounds identified in the analysis of the HELTr via GC-MS have already been described in the reviewed literature on different plant extracts of the *Burseraceae* species, which corroborates the findings of biological tests in other studies, also confirmed in the case of *T. rhoifolia*.

In assays of the biological action of the HELTr in zebrafish, in which, among other effects, the behavioral alterations induced by the compounds of the extract were monitored, highly acceptable patterns were demonstrated, according to the protocols already established [17]. Among them, there is an increased excitability, which results in a loss of posture and habit of resting on the bottom of the aquarium [24], which can occur after exposure to foreign substances, both synthetic and natural [17,70,71].

The reactions observed in zebrafish are possibly caused by the synergistic actions of the compounds present in the HELTr, which are similar to those found in other studies that used the same parameters as those described here.

In this sense, these results were also identified in other studies of zebrafish, in which the animal was treated by immersion with ethanolic extract of *Spilanthes acmella* or with hydroethanolic extract of *Acmella oleracea*, (*A. oleracea*) as an oral treatment [29].

Therefore, it is demonstrated that the HELTr has the property of biological safety, as it is non-toxic. This result is in accordance with other toxicity tests reviewed in the literature, which concluded that *T. rhoifolia* is non-toxic [4,7,8,26]. Likewise, this result for *T. rhoifolia* is in agreement with other studies performed on different species of the Burseraceae family, which also indicated either no or acceptable toxicity after extensive scientific investigations. Thus, the safety of the species studied here is confirmed [4,5,13,14].

In order to enhance the safety of the non-toxic use of the HELTr, histopathology was used, whereby, according to the literature, the level of behavioral changes indicates the internal status of the animal [29]. Hence, histopathological analysis was used to identify internal signs of toxicity.

According to Santos et al. [72] and Ferreira et al. [73], this type of analysis can detect toxicities in specific organs. Several studies report the appearance of histopathological changes in the liver, intestine and kidneys [23,28,72]; thus, the present study evaluated each of these zebrafish organs after HELTr administration.

Regarding the liver, according to Carvalho et al. [74], despite structural differences, the organ is functionally similar to that of mammals. Similarities include the pathways of drug metabolism, as well as lipid and glycogen synthesis and storage [16,75,76]. Thus, in the case of exposure to potentially toxic compounds, the histopathology of the zebrafish liver can be compared to that of mammals because of its physiological conservation [24,77].

The results of the HELTr administration show that the tissue alterations observed in this organ were minimal and, hence, did not affect its normal functioning. Among such changes, cytoplasmic vacuolization was observed in animals treated with the extract, which is widely reported in the literature [16,29] and is associated either with decreased glycogen storage in the hepatocytes or with lipid accumulation. In the present study, the IHC proved that these tissue alterations were within the normal range, which confirms the non-toxicity of HELTr, as already demonstrated by the analyses of behavioral alterations.

Concerning the intestine, it is notable that studies on acute toxicity that evaluate the intestine of zebrafish are extremely important, since this vital organ is the first to be affected by compounds administered orally and, therefore, it is used to evaluate the effects of different substances.

In zebrafish, the intestine is formed by a mucous layer with goblet cells and enterocytes [75], and it is a recycling site for enzymes and macronutrients [78,79]. The sensitivity of this organ to harmful compounds was determined by the evaluation of tissue alterations [75,80]. That said, the infiltration of leukocytes and lymphocytes, as well as the presence of mucus, were observed. These changes were also noticed in a study carried out by Borges [24] and Souza, 2019 [29], in which the authors evaluated the toxicity of the nanoemulsion of *R. officinalis* essential oil and *A. oleracea* extract, respectively. This tissue alteration can be caused by inflammation of the intestinal lamina propria. According to the IHC, the tissue changes observed were not able to compromise the intestine’s functioning, which demonstrates the feasibility of administration and the non-toxic action of the HELTr.

As for the kidney, zebrafish has nephrons composed of the renal corpuscle and proximal and distal convoluted tubules [29], responsible for filtering blood waste and maintaining osmotic balance. This organ is one of the organs most affected by toxic substances [17,30,81].

In this study, the kidney was the organ least affected by the use of the HELTr, which confirms the reports of TPCs from the Brazilian Amazon, who use infusions of this species to treat symptoms of diseases that affect the kidneys [4]. The main tissue alteration observed in this organ was mild tubular hyaline degeneration, characterized by an increase in the number of eosinophilic granules in the cytoplasm of these cells. This tissue alteration occurs due to the reabsorption of the excess protein synthesized in the glomeruli [29], which possibly may not result from the direct action of the HELTr, but from other factors [30,81].

Once the non-toxic action of the HELTr was confirmed, in order to ratify the reports of TPCs [4], zebrafish was adopted as a model animal for prospecting, since it has proved to be experimentally adequate for understanding the acute inflammatory process and for the evaluation of substances and/or extracts with possible anti-inflammatory actions [22,23,24,25,82,83,84].

According to the popular and traditional use of *T. rhoifolia*, which indicates the efficacy and low toxicity of the leaves of the species, the antiedematogenic effect of the HELTr was investigated through the testing of carrageenan-induced abdominal edema in an experimental model of an adult Zebrafish.

Carrageenan was adopted to induce acute inflammation, since this substance produces an edematogenic effect through a biphasic mechanism, mainly characterized by the release of specific pro-inflammatory mediators in different periods. In the initial phase, after carrageenan injection (0–1 h), histamine and/or serotonin are released, which are responsible for vasodilation and increased vascular permeability at the beginning of the inflammatory process. The second phase (1–6 h), mediated mainly by the release of prostaglandins, is characterized by the presence of cellular infiltrate, composed of macrophages, eosinophils and, mainly, lymphocytes. Both phases can be mediated by the release of kinins [85,86].

Edema is a common sign of acute inflammation, and its formation is caused by vasodilation and increased vascular permeability due to the effects of inflammatory mediators, which enable an increase in the plasma flow, protein extravasation and edema formation by exudate accumulation in the extracellular matrix [27].

Under these mechanisms, induced in a carrageenan-induced abdominal edema assay, the HELTr showed an anti-edematogenic effect, with high levels of effectiveness and a substantially long duration (Figure 3). This result indicates that the antiedematogenic action mechanism of the HELTr may involve the serotonin and/or histamine pathways, as well as the kinins and/or prostaglandin pathways [87]. The HELTr inhibited edema at all times throughout the experiment (Figure 3), demonstrating a similar effect to the standard drug (indomethacin) and being pharmacologically more potent than the former at the maximum peak of the edema (5th hour) (Figure 5). The data suggest that the HELTr produced a pharmacological synergism, acting through a potent anti-inflammatory effect, possibly through the inhibition of the synthesis and/or release and/or action of inflammatory mediators involved in acute inflammation [87].

According to the literature, the potential anti-inflammatory effects of the HELTr and its pharmacological synergism probably derive from hexadecanoic acid, ethyl ester [56,57]; propanoic acid, ethyl ester [37,38,58,59], *n*-hexadecanoic acid, benzophenone [35,36], and acetic acid ethenyl ester [20,66]; and 3-hydroxybenzoic acid [15,69]. Furthermore, there are extensive chemical and pharmacophoric studies of other *Burseraceae* species that also prove their anti-inflammatory properties [4,60,61]. It is believed that these compounds may play an important role in inhibiting the inflammatory process, interfering with the mechanisms triggered by carrageenan. It is therefore suggested that the significant anti-inflammatory activity of the HELTr is probably a result of the presence of these compounds, with a possible synergistic effect.

The HELTr, traditionally used for the treatment of inflammatory processes and/or diseases, also showed an anti-edematogenic effect at all the doses tested, as demonstrated in this study. Other studies involving the hydroethanolic extract of the leaves of *Commiphora leptophloeos, Protium kleinii* and *Boswellia serrata (“salai guggal)”*—species of the *Burseraceae* family—showed antiedematogenic results in zebrafish and rodents in the context of edema induction by carrageenan [88,89,90]. The species *T. rhoifolia* [4,21] and *C. album* [60] exhibited potent anti-inflammatory activities in previous studies on different in vivo and in vitro models [4,59,60,61].

According to Carvalho et al. [74], the injection of an inflammatory agent (such as carrageenan) into the abdominal region of zebrafish can provoke reactions in vital organs, especially the liver, intestine and kidneys, given the small size of the animal. The authors also state that the liver is the vital organ for the detoxification process of substances, and any dysfunction of its tissue can be harmful to the animal or even cause death [91,92,93,94]. Hyperemia, an alteration observed in zebrafish in the negative control group (intraperitoneal carrageenan and DMSO, p.o.), occurs in an attempt to increase the general blood flow in the liver and the release of nutrients and oxygen to the affected areas, preventing hypoxia [29,30,91,95,96,97].

The zebrafish’s metabolism is extremely agile and surpasses those of other Teleost species. This fact may help to explain the intense vacuolization of the cytoplasm of hepatocytes, which, in some cases, indicates the beginning of a degenerative process that can result in the dysfunction of metabolic processes after exposure to toxic substances [81].

Studies evaluating the anti-inflammatory activity of substances report the presence of histopathological changes in the zebrafish intestine after intraperitoneal injections of carrageenan [24,25,98]. Previously, intestine studies were based only on informal observations [79].

In this study, the vacuolization of enterocytes and the infiltration of leukocytes and lymphocytes were recorded. According to Carvalho et al. [74], vacuolization is a type of alteration that can compromise the body’s ability to absorb nutrients. According to Borges [24], the presence of these alterations may be related to the invasive technique of intraperitoneal injection, which can damage the intestine. Invasive procedures in fish can generate inflammation in the intestinal lamina propria and cause the infiltration of leukocytes into the epithelial tissue of the intestine [74], in addition to causing an increase in the number of defense cells [79]. The zebrafish kidney contains nephrons responsible for filtering waste from the blood and capturing salt and water. It includes the lymphoid and hematopoietic regions, as well as steroidogenic and endocrine cells [24]. According to Holden et al. [80], the main role of the kidneys in freshwater Teleost fish is to eliminate the large volume of water that enters through the mouth, not to store it.

Finally, according to Carvalho et al. [74], tubular changes in the zebrafish kidney may occur due to metabolic dysfunctions caused by toxic compounds. These changes can eventually induce renal necrosis if the toxicity is high enough [30]. In this study, tubular changes, tubular degeneration and tubular cell hyperplasia were mild, with no necrosis or tissue dysfunction.

Based on the data gathered, molecular docking studies were performed, which enabled the evaluation of potential COX-2 inhibitors through assessing the similarity of interactions with amino acid residues in the binding site [98]. The controls showed significant stability and anti-inflammatory actions; however, studies point to their adverse effects when administered [52,99,100]. The molecules hexadecanoic acid, ethyl ester, benzophenone and n-hexadecanoic acid, identified in HELTr, showed good profiles for the inhibition of the COX-2 enzyme, as the interactions established at the receptors used in the docking study were similar to the controls (RCX, IMN and CEL).

## 4. Materials and Methods

### 4.1. Plant Material

The plant material, consisting of leaves, was collected in January 2021 at the production center for seedlings and seeds of the Environmental Police Battalion (BPA) of Candeias do Jamari, Rondônia, Brazil, with the geographic coordinates 08°48′35″ S and 62°41′44″ W. The samples were identified and deposited in the herbarium of the Federal University of Rondônia, marked as belonging to the species *T. rhoifolia*, under the deposit number 10.242/2015. The authorization for the transport and use of plant material was issued by the Chico Mendes Institute for Biodiversity Conservation, ICMbio/IBAMA, at SISBio/Brazil, under the registration number 57,314 (with property’s general registration IBAMA/CGEN 5571551).

### 4.2. Preparation of HELTr

The leaves were dried in a circulating air oven at 38 ± 2 °C for 5 days, pulverized in a portable knife mill (Oster 4126, São Paulo, Brazil) and subjected to extraction with 96° PA ethanol (Vetec Química, Xerém, Rio de Janeiro, Brazil). For each extract, the proportions of 50 g of *T. rhoifolia* leaves for 60 mL of solvent were used, renewed after 12, 24, 36 and 48 h. Each cycle was submitted to an ultrasound bath (Elmasonic P, Elma, Singen, Germany) with a power setting of 80 W, frequency of 37 Hz and the temperature at 25 °C for 15 min, with a 12 h interval, in order to maximize the extraction of the compounds by percolation. Then, the solutions were filtered through qualitative filter paper, following the methodologies adapted in other experiments [81,82]. The extracts were dried in a rotary evaporator (Fisatom, Brazil) at a temperature below 38 °C. Afterwards, the extracts were lyophilized (Marconi Freeze Dryer, Brazil) for 72 h and stored, being hermetically closed under refrigeration, resulting in masses of residual extracts, as leaves of 9 g, called HELTr.

### 4.3. GC-MS Analysis

Chemical analyses of the HELTr samples were performed at the EST-UEA Chemical Analysis Center using a gas chromatograph (GC, model 7890B, Agilent Technologies, Inc., São Paulo, Brazil) coupled with a quadrupole mass spectrometer (EM, model 5977A, Agilent Technologies, Inc. MSD), using a DB5-MS 30 m × 250 µm × 1 µm MDS capillary column. When injecting the samples into the GC, the injector operated in split mode (10:1 ratio) at a temperature of 280 °C, with a column flow rate of 12 mL·min^−1^ and He as the carrier gas. The initial GC temperature was 25 °C, followed by isothermal treatment for 3 min and successive increases by 5 °C·min^−1^, giving a total of 53 min until it reached 290 °C, when it remained isothermal for 4 min [84]. The total time of analysis during the entire process was 60 min for each sample. The mass spectrometer was operated through the monitoring of all ions of a mass/charge ratio (*m*/*z*) between 40 and 500. The temperature of the electron source for the mass spectrometer was 290 °C and the quadrupole temperature was 150 °C, without the need for quantification and with exploratory analysis [93]. The identification of the compounds was performed through assessing the similarity of the experimental mass spectra with those of the Nist Research Library (similarity values of the spectra being greater than 80%).

### 4.4. Study of the HELTr in Adult Zebrafish

#### 4.4.1. Experimental Animals

The animals used—adult zebrafish—were provided by the company Power Fish Pisciculture, located in Itaguaí-RJ, Brazil. Males and females of the wild AB strain were used, which were kept in aquariums at the Zebrafish Platform of the Pharmaceutical Research Laboratory of the Federal University of Amapá (UNIFAP-Brazil) over an adaptation period (40 days). A 12 h circadian rhythm was observed (light period from 7:00 a.m. to 7:00 p.m.), with a controlled temperature (23 ± 2 °C) and feeding based on commercial flake feed (Alcon Colors, Santa Catarina, Brazil) twice a day, in a monitored environment, following the parameters of pH (6.0–8.0), conductivity (8.2 ± 0.2) and cleaning of the recirculation system [29]. The experiments were authorized by the Ethics Committee for Animal Use of the Federal University of Amapá (Brazil), registration number 002/2022/unifap.

#### 4.4.2. Study of Acute Oral Toxicity

The assessment of the acute toxicity of the HELTr in adult zebrafish was determined by the threshold test, following the parameters of the Organization for Economic Cooperation and Development (OECD), 425 OECD [29], with adaptations. The toxicity test consisted of the use of oral doses [95]. The animals were divided into control and treatment groups (5 animals/group), fasted for 24 h, and then weighed and treated orally according to the methods described in [96]. The HELTr was administered at a dose of 2000 mg/kg, orally, by the gavage method, with a volumetric pipette (HTL Lab Solutions Co., São Paulo, Brazil). The volume was calculated according to the weight of the animal [25,29]. The animals were temporarily immobilized through thermal anesthesia, in water at a temperature of 4 ± 2 °C in a low-density, center-concave damp sponge measuring 11 cm × 7.5 cm × 2.3 cm, which was prepared for this purpose [29].

#### 4.4.3. Behavioral Analysis and Mortality

After the gavage procedure, the animals were observed for behavioral changes, classified as: stage I—increase in swimming activity, spasms and tail tremors; stage II—circular swimming and loss of posture; and stage III—clonus, loss of motility, animal resting at the bottom of the aquarium and death [17]. In the absence of a response to mechanical stimulation and the absence of operculum movement, the animal was considered dead [17,29]. At the end of the observations, the animals were euthanized under anesthetic cooling, in accordance with the Veterinary Medical Association’s American Guidelines for Animal Euthanasia [74,95].

#### 4.4.4. Evaluation of the Anti-Inflammatory Activity of the HELTr

Zebrafish are sensitive animals. Therefore, to meet the principles of the “3RS” of bioethics in the carrageenan-induced abdominal edema test, a pilot optimization test was carried out in order to evaluate the resistance of the animals to the handling process during weighing.

In this step, adult zebrafish (*n* = 9) were randomly selected, weighed separately and injected intraperitoneally with carrageenan (300 µg, 20 µL). The average weight of the animals ranged between 200 and 500 mg. Every hour, from 1 to 5 h after the carrageenan injection, the animals were individually weighed. From the results, it was observed that animals weighing >300 mg showed greater resistance to manipulation at the time of weighing, remaining alive until the end of the 5 h period of evaluation. Thus, in this study, the use of animals weighing >300 mg was recommended.

#### 4.4.5. Treatment Groups

To assess the potential anti-inflammatory effect of the HELTr, we used the methodology described by Ekambaram et al. [22], with adaptations. The animals were selected through the initial screening, as described in Section 4.4.4., and divided into six groups (*n* = 12/group). The assays were performed in triplicate. The HELTr doses used in this experiment were established based on the acute oral toxicity assay. The groups were treated with different substances as follows: control group (PBS + SS)—animals treated with saline solution (2 μL, p.o.) and PBS, a substance used to solubilize carrageenan (20 μL, i.p.); negative control group (DMSO + CAR)—animals treated with DMSO (2 µL, p.o.) and carrageenan (300 µg, 20 µL, i.p.); positive control group (indomethacin)—animals treated with indomethacin (10 mg/kg, p.o., Sigma Co. São Paulo, Brazil) and carrageenan (300 µg, 20 μL, i.p.), HELTr at doses of 100, 200 and 500 mg/kg (p.o.) and carrageenan alone (300 µg, 20 µL, i.p.).

#### 4.4.6. Induction of Abdominal Edema

Carrageenan (300 µg, iota type II, Sigma Co., Lot 65H1096, São Paulo, Brazil) was used to induce abdominal edema in the adult zebrafish, according to the methodology validated by Huang et al. [73], with adaptations. Intraperitoneal (i.p.) injection was adopted as the method of administration for inducing inflammation in the adult zebrafish, in order to ensure that the exposure concentrations were within the target values [87]. For the injections (i.p), a short needle, U-30 BD Ultra-Fine™ 3/10 mL, and 8 mm (5/16″) × 31G insulin syringes (Becton Dickinson and Company, NJ, USA) were used, following pre-established protocols [88].

The animals were anesthetized in water at a temperature of 10 ± 2 °C and gently placed on a damp sponge of a 1 kg/m^3^ density, center-concave, with a 0.5 cm radius and dimensions of 3 cm in height, 7 cm in width and 4 cm in length. With the abdomen facing upwards, the needle was inserted parallel to the spine into the midline of the abdomen, posterior to the pectoral fins (the pectoral fins were used as a reference point). The injection procedure (i.p.) was conducted in order to ensure that the animal did not remain out of the water for more than 10 s. After injection, animals were placed in a separate tank with system water (25 ± 2 °C) to facilitate their recovery from anesthesia. All animals that recovered within 2 to 3 min after injection continued to be used in the experiment, while animals that did not recover during this period were discarded. For the analysis of the abdominal edema, after carrageenan injection (i.p.), the animals were individually weighed and their body weight was measured every 1 h from 1 to 5 h.

### 4.5. Histopathological Analyses

For the histopathological analyses of the liver, intestine and kidneys, the animals were fixed in Bouin’s solution for 24 h and decalcified in EDTA solution (ethylenediamine tetra acetic acid, Sigma Co., São Paulo, Brazil) for 48 h. The samples were dehydrated in alcohol series (70, 80, 90 and 100%), diaphanized in xylene and embedded in paraffin. Samples were sectioned at 5 µm using a microtome (Brand Rotary Microtome Cut 6062, Slee Medical, Germany), and the histopathological analysis was performed after tissue sections were stained with hematoxylin and eosin, as described by Souza et al. [17]. The images were analyzed using an Olympus BX41-Micronal Microscope and photographed with a MDCE-5C USB 2.0 digital camera.

#### Assessment of Histopathological Changes

The IHC was calculated from the levels of tissue changes observed in the liver, intestine and kidneys. The changes were classified into levels I, II and III, and the IHC value was used to indicate whether the organ was healthy (0 to 10), had mild to moderate changes (11 to 20) or moderate to severe (21 and 50) damage, or was irreversibly damaged (>100) [100,101,102]. Thus, the indices were calculated according to the following equation:$$\frac{I=\sum_{i-1}^{na}ai+10\;\sum_{i-1}^{nb}bi+102\;\sum_{i-1}^{nc}ci}{N}$$

### 4.6. Statistical Analysis

The statistical estimation of the significance of the results obtained in the trials was performed using the ANOVA test, in which the objective was to compare the means between the control and treated groups, establishing significant probability values (*p*) < 0.05 [99] and considering the highly significant values of *p* < 0.01.

In reference to studies involving zebrafish, a one-way analysis of variance (ANOVA) was used, as well as Dunnett’s multiple comparison test. The IHC was established through statistical analyses and pre-established parameters were determined by one-way ANOVA (Kruskal–Wallis) and the Student–Newman–Keuls test. For this purpose, values of *p* < 0.05 were adopted as statistically significant values. The data obtained were expressed as mean ± standard error of the mean. Graphs were generated and recorded using the GraphPad Prism 6.0 software. The analyses of the results of the anti-inflammatory activity assessments were performed using GraphPad Prism software version 5.0, using the one-way ANOVA test followed by the Tukey–Kramer test, in order to obtain better comparisons between the tested and control groups. The values established at *p* < 0.05 and *p* < 0.01 were considered statistically significant. Data were presented as mean ± standard error of the mean [99].

### 4.7. Molecular Docking Simulations

#### 4.7.1. Enzyme and Inhibitor Structure Selection

The structures of the enzyme COX-2, elucidated by X-ray diffraction, were downloaded from the Protein Data Bank (PDB) with PDB ID: 5KIR (*Homo sapiens*) and 4COX (*Mus musculus*), complexed with Rofecoxib (RCX) and indomethacin (IMN) at resolutions of 2.7 and 2.9 Å, respectively [103,104]. RCX, IMN and Celecoxib (CEL) were used as control ligands in the molecular docking studies based on established standard protocol [52,104,105,106].

#### 4.7.2. Docking Study

The docking study was performed using AutoDock 4.2/Vina 1.1.2 via Graphical Interface PyRx (Version 0.8.30), accessed on 20 August 2022. The assignment of the protonation and tautomeric states of the ligands was performed using the Discovery Studio^®^ 2.0 program (2008), while the hydrogen atoms of the proteins were added with PROPKA using pH 7. In the COX-2 docking study, the specific ligands complexed in AutoDock 4.2/Vina 1.1.2 and PyRx 0.8.30 were used (https://pyrx.sourceforge.io, accessed on 20 August 2022). The validation of the molecular docking protocols was performed by overlapping the crystallographic ligand (experimental pose) with the best conformation obtained (docking pose) based on the root mean square deviation (RMSD) value [47,50,52,106].

## 5. Conclusions

Our results validate the main narratives of the TPCs of the Brazilian Amazon, who use the species as an anti-inflammatory and indicate the non-toxicity of *T. rhoifolia*. Furthermore, the chemical profile of the HELTr, analyzed via GC-MS, had a high degree of similarities to the identified compounds, so that there was no significant difference between the chemical profiles of the species studied and the reports in the literature. However, this topic still needs further study and new tests in order to evaluate the anti-inflammatory activity with respect to different parts of the plant and fractions or groups of substances present in the species *T. rhoifolia*.

The HELTr has an acceptable degree of safety for acute toxicity based on the behavioral change, mortality and histopathology analyses. HELTr showed a significant anti-inflammatory action in zebrafish at all the doses tested, and this action was confirmed by molecular docking studies, in which the molecules A, B and I, identified in HELTr by GC/MS, showed good profiles for COX-2 enzyme inhibition. Thus, the secondary metabolites present in the species are demonstrated to have a high pharmacological potential, which adds new value to our knowledge of an endemic species not known to the industry. The chemical profile of HELTr had a high degree of similarity with the suggested compounds via GC-MS, so that there was no significant difference between the chemical profiles of the studied species and the literature reports.

HELTr exhibited an acceptable degree of safety for acute toxicity according to the analyses of behavioral changes, mortality, and histopathology. HELTr exhibited significant anti-inflammatory action in zebrafish at all the doses tested, and this action was confirmed by molecular docking studies, in which molecules A, B and I, identified in HELTr by GC/MS, showed good profiles for COX-2 enzyme inhibition. However, with regard to the experiments, other mechanisms of inflammatory induction should be tested at different doses and using different fractions or compounds of the extract of the species studied for a better understanding of the profile and potential of secondary metabolites present in the species.

Our results validate the main narratives of the TPCs of the Brazilian Amazon regarding the traditional practices, which attribute the property of anti-inflammatory activity to the species *T. rhoifolia* and indicate the toxicity of the plant. Furthermore, it was demonstrated that the secondary metabolites present in this species have high pharmacological potential, adding new value to this endemic species, which is not known to the industry.

## Figures and Tables

**Figure 1 molecules-27-07741-f001:**
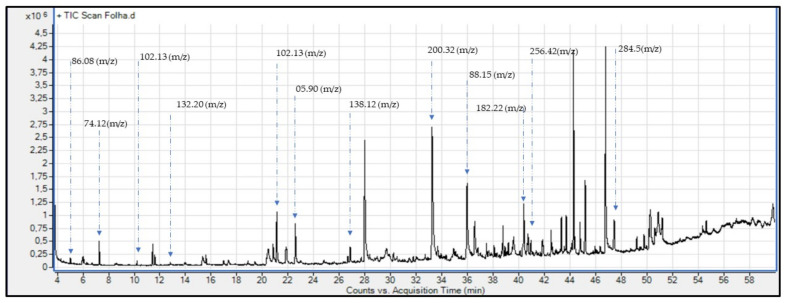
TIC of the HELTr according to molar mass indications, in order of retention time.

**Figure 2 molecules-27-07741-f002:**
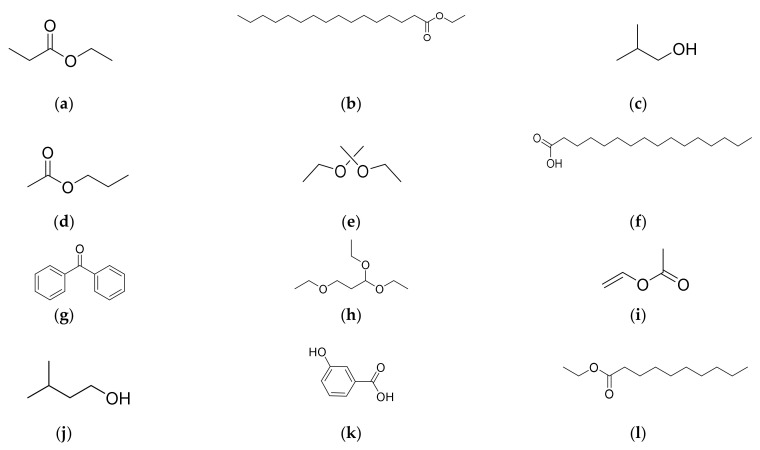
Main chemical compounds identified in the HELTr by GC-MS through TIC, NITS/MS Library. In order of retention time: (**a**) propanoic acid, ethyl ester; (**b**) hexadecanoic acid, ethyl ester; (**c**) 1-propanol 2-methyl; (**d**) *n*-propyl acetate; (**e**) propane, 2,2-diethoxy; (**f**) *n*-hexadecanoic acid; (**g**) benzophenone; (**h**) propane,1,1,3-triethoxy; (**i**) acetic acid, ethenyl ester; (**j**) 1-butanol, 3-methyl; (**k**) 3-hydroxybenzoic acid; (**l**) decanoic acid, ethyl ester.

**Figure 3 molecules-27-07741-f003:**
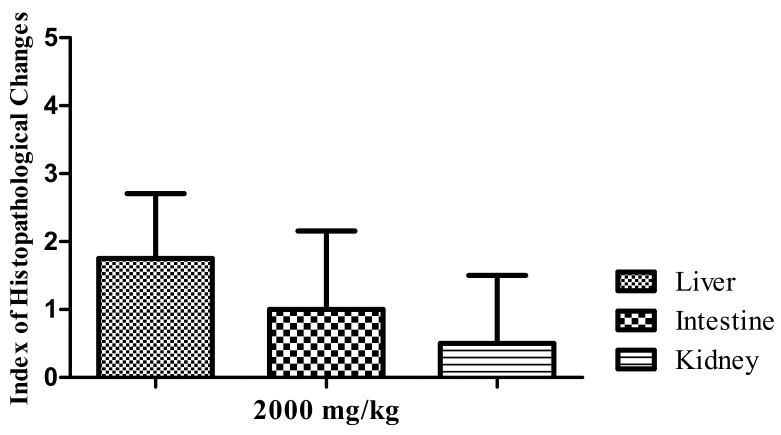
IHC in the liver, intestine and kidneys of adult zebrafish in the acute oral toxicity test performed using the HELTr 2000 mg/kg dose. Data show the mean ± SD (*n* = 5/group). Statistical analysis was performed through one-way ANOVA followed by the post hoc Tukey test.

**Figure 4 molecules-27-07741-f004:**
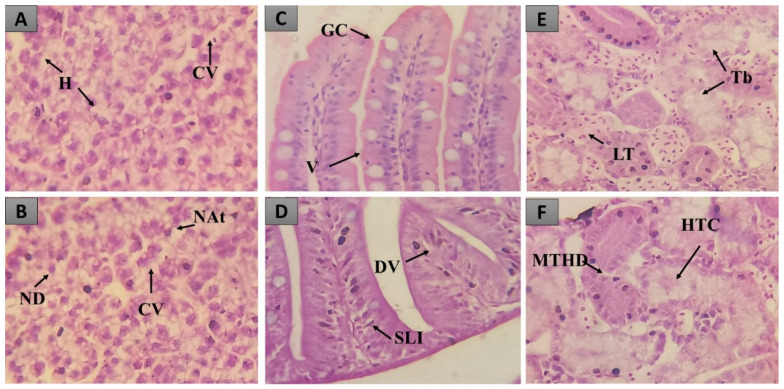
Histopathological changes observed in the liver, intestine and kidneys of zebrafish treated with the HELTr 2000 mg/kg dose. In (**A**,**B**), there are observed normal hepatocytes in the liver tissue (H), cytoplasmic vacuolization (CV), nuclear atypia (NAt) and nuclear degeneration (ND). In (**C**,**D**), intestinal tissue with normal goblet cells (GC), villi (V), the dilation of the vessels present in the villi (VD) and stromal leukocyte infiltration (SLI) can be seen. In (**E**,**F**), renal tissue with normal tubules (Tb) and lymphoid tissue (LT), mild tubular hyaline degeneration (MTHD) and tubular cell hyperplasia (HTC) can be observed. H&E Staining.

**Figure 5 molecules-27-07741-f005:**
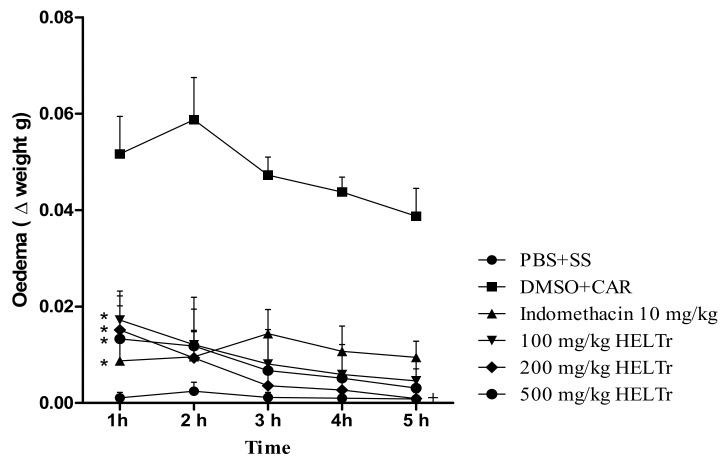
Effect of the HELTr on the inhibition of carrageenan-induced abdominal edema in adult zebrafish. Effects of the HELTr (100–200–500 mg/kg, oral administration, p.o.) and indomethacin (10 mg/kg, p.o.) doses or of their vehicle (5% DMSO, 2 µL/animal, p.o.) throughout the experimental period (0–5 h) on carrageenan-induced edema (300 µg/animal, 20 µL, intraperitoneal—i.p) and/or phosphate-buffered saline—PBS (20 µL/animal, i.p.). Data are expressed as mean ± standard error of mean (*n* = 12/group). * Probability *p* ≤ 0.05 compared to the negative control (DMSO + CAR), + *p* ≤ 0.05 vs. indomethacin 10 mg/kg. Statistical analysis was performed through one-way ANOVA followed by the post hoc Tukey test.

**Figure 6 molecules-27-07741-f006:**
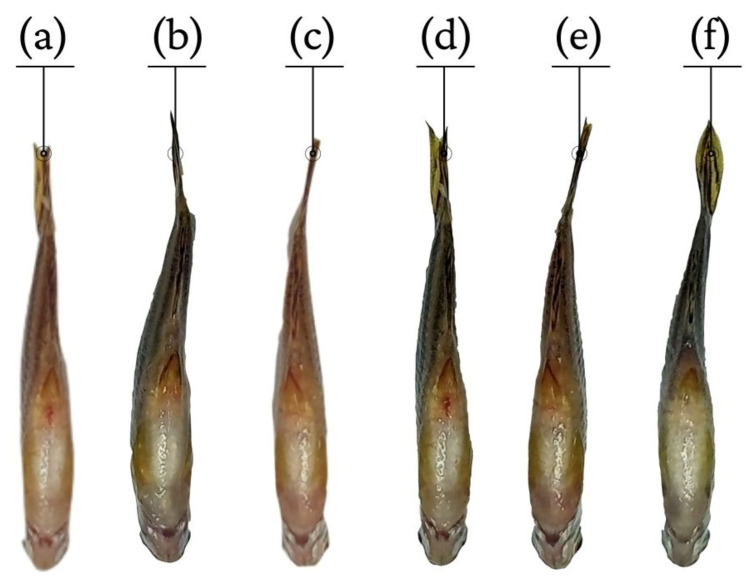
Macroscopic view of the effect of the oral administration of the solution: (**a**) PBS—2 μL; (**b**) carrageenan—200 µg/anima; (**c**) indomethacin—10 mg/kg); (**d**), (**e**) and (**f**) HELTr—100, 200 and 500 mg/kg, respectively.

**Figure 7 molecules-27-07741-f007:**
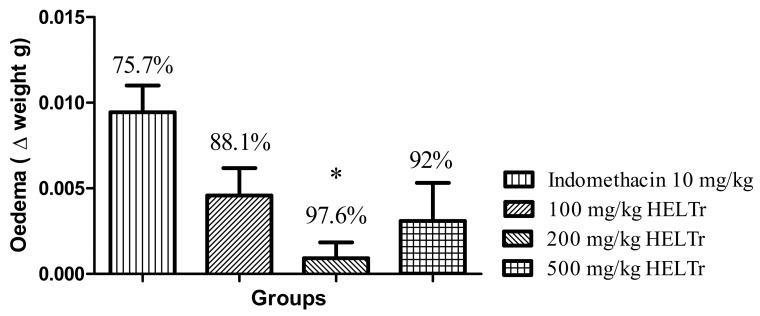
Effect of the HELTr on the inhibition of carrageenan-induced abdominal edema in adult zebrafish at the maximum edema peak (0–5 h). Each column represents the mean ± standard error of the mean (*n* = 12/group). The numbers above the bar indicate the percentage of inflammation inhibition for the positive control (indomethacin 10 mg/kg) and HELTr (100–200–500 mg/kg, p.o.) groups. Statistical analysis was performed through one-way ANOVA followed by the post hoc Tukey test. (* *p* ≤ 0.05 control (indomethacin) vs. 200 mg/kg of the HELTr).

**Figure 8 molecules-27-07741-f008:**
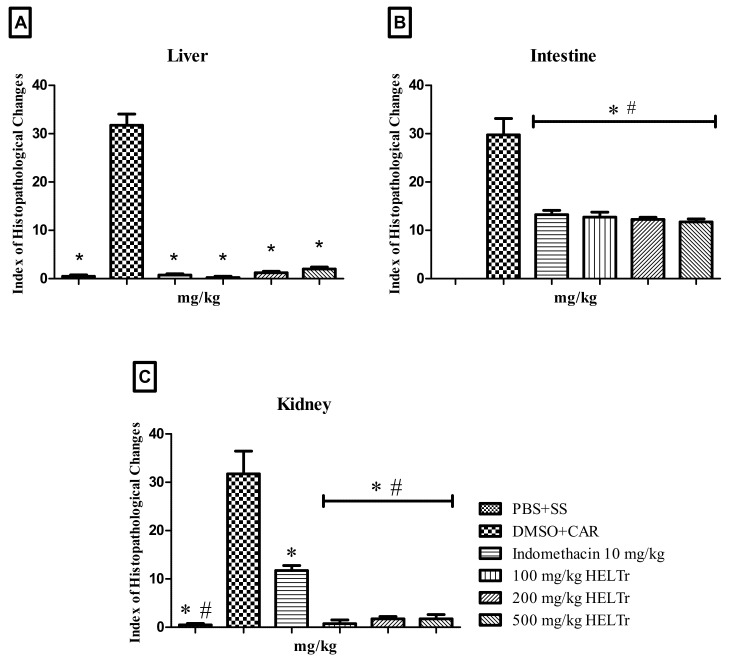
IHC in the liver (**A**), intestine (**B**) and kidneys (**C**) of adult zebrafish in the evaluation of the effects of the control (saline and DMSO, p.o.) and HELTr (100–200–500 mg/kg, p.o.) groups on histopathological changes in the carrageenan-induced abdominal edema test. * Probability *p* ≤ 0.05 compared to the control (PBS+SS), # *p* ≤ 0.05 vs. control negative (DMSO+CAR). Data are expressed as mean ± standard error of mean (*n* = 12/group). Statistical analysis was performed through one-way ANOVA followed by the post hoc Tukey–Kramer test.

**Figure 9 molecules-27-07741-f009:**
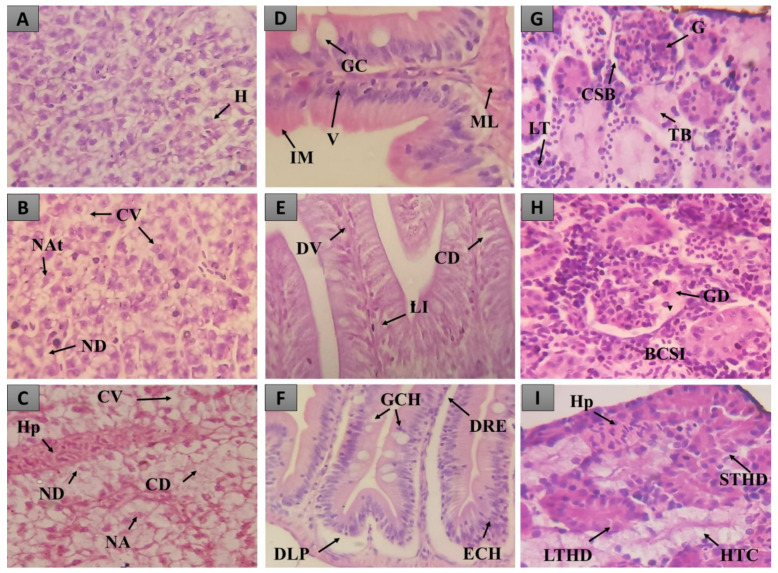
Histopathological changes observed in the liver, intestine and kidneys of adult zebrafish in the evaluation of the effects of control (saline solution and DMSO, p.o.) and HELTr (100–200–500 mg/kg, p.o.) groups. In (**A**–**C**), liver tissue with normal hepatocytes (H), cytoplasmic vacuolization (CV), nuclear atypia (NAt), nuclear degeneration (ND), cellular degeneration (CD), nuclear atrophy (NA) and hyperemia (Hp) are observed. In (**D**–**F**), the intestinal tissue is distinguished by normal goblet cells (GC), villi (V), intestinal mucosa (MI), muscular layer (ML), dilatation of vessels present in the villi (VD), leukocyte infiltration (LI), goblet cell hyperplasia (GCH), epithelial cell hypertrophy (ECH), displacement of lamina propria (DLP) and detachment of the epithelial lining (DRE). In (**G**–**I**), renal tissue with normal tubules (Tb), glomerulus (G), Bowman’s capsule space (BSC), lymphoid tissue (LT), glomerular degeneration (GD), increase in Bowman’s capsule space area (ISBR), hyperemia (Hp), mild (MTHD) and severe (STHD) tubular hyaline degeneration and tubular cell hyperplasia (CTH) can be seen. H&E Staining.

**Figure 10 molecules-27-07741-f010:**
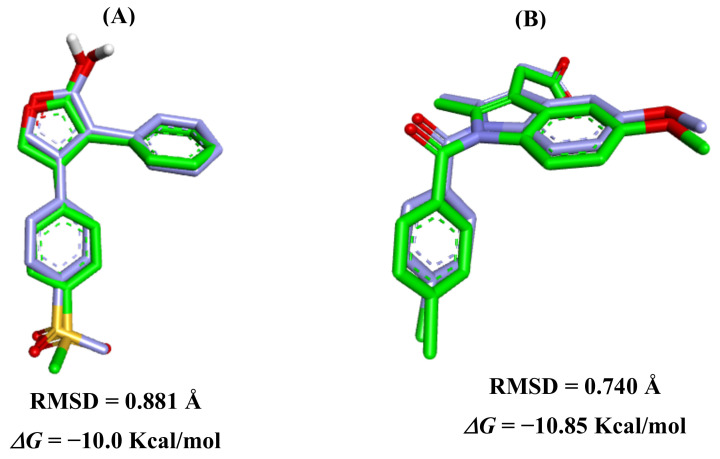
Overlays of crystallographic ligand poses (purple) with docking poses (green): (**A**) RCX for *Homo sapiens* (PDB ID: 5KIR) and (**B**) IMN for *Mus musculus* (PDB ID: 4COX).

**Figure 11 molecules-27-07741-f011:**
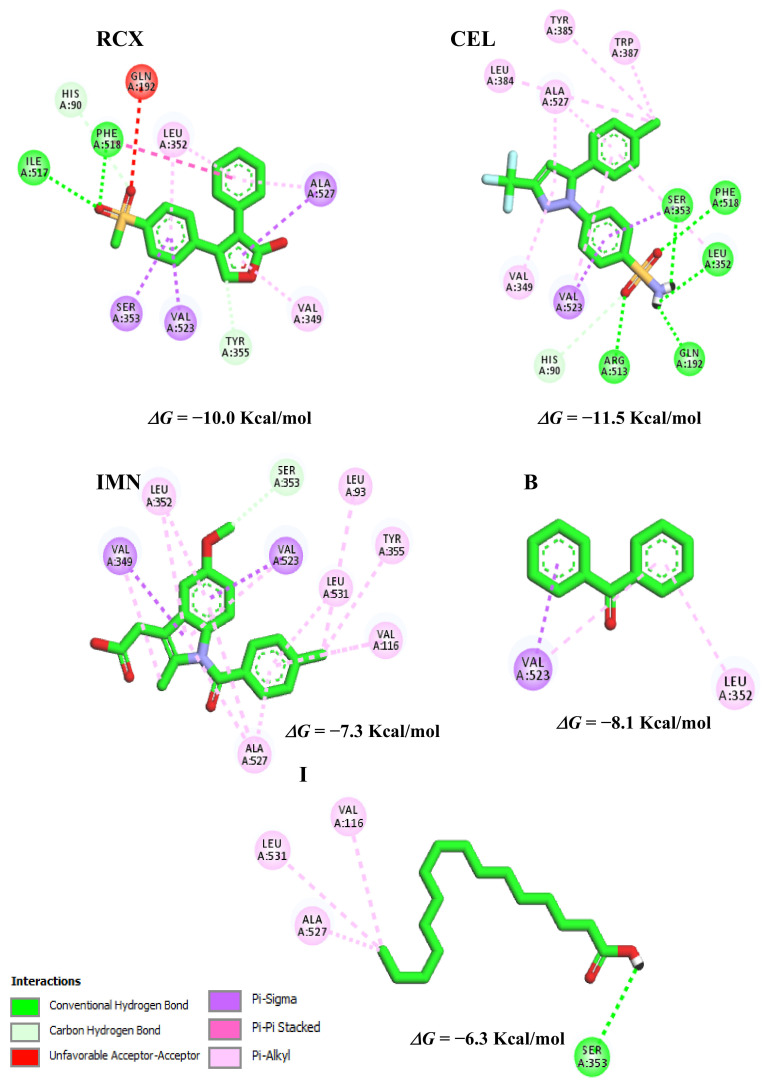
Interactions of the controls, RCX, CEL, IMN, and the potential inhibitors B and I with the COX-2 binding site (PDB ID: 5KIR).

**Figure 12 molecules-27-07741-f012:**
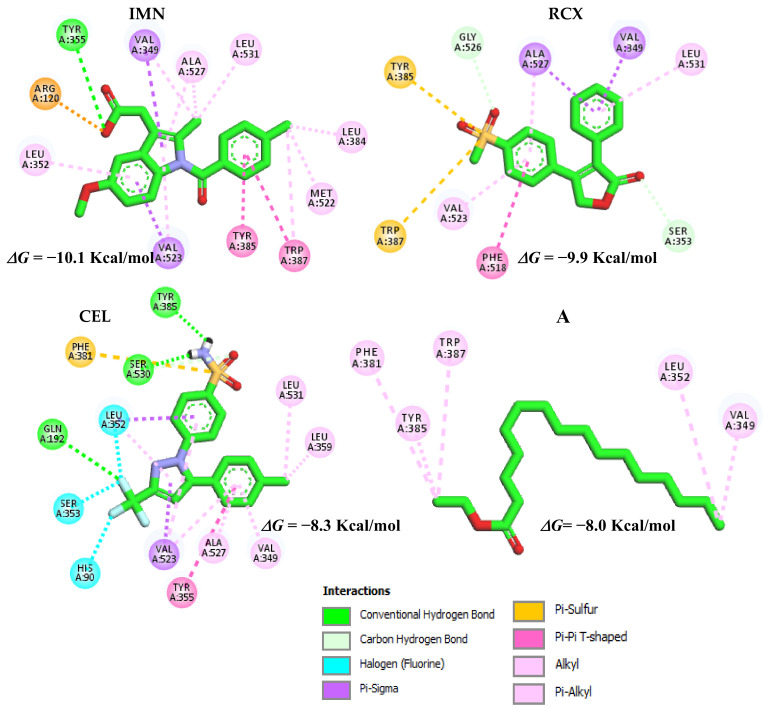
Interactions of IMN controls, RCX, CEL and potential inhibitor A with the COX-2 binding site (PDB ID: 4COX).

**Table 1 molecules-27-07741-t001:** Main chemical compounds of the HELTr identified with GC-MS.

Names of Compounds	Molecular Formula	Similarity	RT *	Molar Mass *(m*/*z)*	Class of Metabolites
Propanoic acid, ethyl ester	C_5_H_10_O_2_	98.0	10.41	102.06	Ester
Hexadecanoic acid, ethyl ester	C_18_H_36_O_2_	95.4	47.93	284.5	Fatty acid
1-Propanol 2-methyl	C_4_H_10_O	95.2	07.93	74.12	Alcohol
*n*-Propyl acetate	C_5_H_10_O_2_	95.0	20.61	102.13	Ester
Propane, 2,2-diethoxy	C_7_H_16_O_2_	94.0	12.69	132.20	Acetone
*n*-Hexadecanoic acid	C_16_H_32_0_2_	90.9	41.91	256.42	Fatty acid
Benzophenone	C_13_H_10_O	90.0	40.38	182.22	Acetone
Propane,1,1,3-triethoxy	C_9_H_20_O_3_	89.6	23.87	176.25	Ester
Acetic acid ethenyl ester	C_4_H_6_O_2_	80.0	05.90	86.08	Ester
1-Butanol, 3-methyl	C_5_H_12_O	80.0	36.0	88.15	Alcohol
3-Hydroxy, benzoic acid	C_7_H_6_O_3_	71.2	27.97	138.12	Benzoic acid
Decanoic acid, ethyl ester	C_12_H_24_O_2_	70.3	33.64	200.32	Ester

RT *: retention time, in seconds.

**Table 2 molecules-27-07741-t002:** Effects of the treatments with different oral doses of the HELTr on the behavioral reactions of zebrafish.

Group	Stage I	Stage II	Stage III	Total	%
2000 mg/kg	1/3	0/2	2/4	3/9	33.3
Control (DMSO)	1/3	0/2	1/4	2/9	22.2

**Table 3 molecules-27-07741-t003:** Oral bioavailability of the molecules (RO5).

Molecules	MW ^a^	CLogP ^b^	HBA ^c^	HBD ^d^	Ro5 ^e^
Normal range	<500	<5	<10	<5	Max.4
RCX	314.4	2.3	4	0	0
IMN	357.8	4.3	4	1	0
CEL	381.4	3.4	7	1	0
B	182.2	3.4	1	0	0
I	256.4	6.4	2	1	1
A	284.5	7.8	2	0	1

^a^ molar weight; ^b^ the predicted skin permeability; ^c^ number of hydrogen bonds accepted by the molecule; ^d^ number of hydrogen bonds donated by the molecule; ^e^ number of violations of Lipinski’s “rule of five”.

## Data Availability

Not applicable.

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
