# Peer review of "Acute Toxicity and Anti-Inflammatory Activity of Trattinnickia rhoifolia Willd (Sucuruba) Using the Zebrafish Model"

_molecules, 2022, doi:10.3390/molecules27227741_

Round 1
Reviewer 1 Report
"Trattinnickia rhoifolia Willd (“sucuruba”): Study of Acute Toxicity and Anti-Inflammatory Activity Using Zebrafish” by de Souza et al. seems to be quite informative. However, the authors must revise the manuscript in accordance with the comments below if they desire it to be considered for publication in the Molecules Journal.
1. The authors should first consider about altering the title to the one stated below.
“Acute Toxicity and Anti-Inflammatory Activity of Trattinnickia rhoifolia Willd (Sucuruba) Using Zebrafish Model”
2. Abstract: Rewrite the abstract with care. The study overall content is not correctly reflected in the abstract. Although it is lengthy, certain important details are missing.
For one statement, line 22–26 is excessively long. Make it short.
Line 26 fails to convince in any manner why acute toxicity and anti-inflammatory activity of Trattinnickia rhoifolia Willd (Sucuruba) has been carried out? The research gap is not clearly mentioned.
It is not necessary to list all the compound names in the abstract in lines 34–37.
Additionally, future directions for this study is to be provided in the abstract's conclusion section.
3. Introduction: Once again, it is excessively long and essential information’s are lacking. I couldn’t locate the specific research gap indicating clearly on why Trattinnickia rhoifolia has been selected in the present study for anti-inflammatory activity. The rationale behind why natural anti-inflammatory molecules are necessary and what are the drawbacks of the currently available NSAID’s to be provided. I recommend author to refer the following article https://doi.org/10.3390/molecules27030734
4. In lines 88 to 94, please use this latest article as a reference to discuss the advantages of using the zebrafish model over other animal models. https://doi.org/10.3390/molecules27082572
5. Line 164 – change the word “Lower than 10” to below 3. Since the values are lower than 3 in figure 3.
6. Figure 4 and 9 legend – Include the magnification (For example 100X).
7. Line 219 – (o.a.) to be changed as (p.o.). since standard format for oral administration is (p.o.) only. This is to be corrected throughout the manuscript.
8. Figure 5, 7 and 8 – In the figure bar diagram indicated as 100 mg/kg Leaves, 200 mg/kg Leaves and 500 mg/kg Leaves, this is to be changed as 100 mg/kg HELTr, 200 mg/kg HELTr and 500 mg/kg HELTr. This also should be uniform throughout the manuscript.
9. The discussion section has too many short paragraphs and it is too long. I had a hard time paying attention to it. I had a hard time paying attention to it. Some of the results was repeated in the discussion. To help readers grasp what the authors are truly discussing about here, I advise authors to cut out the repetition and keep their sentences concise.
10. Section 4.1 – Change “Botanical Material” to “Plant Material”
11. Section 4.2 – Change HELTr Obtainment to Preparation of HELTr
12. Line 757 – Include the accessed date of the docking software/website
13. Line 763 – PCTs, but in other places of the manuscript, it is mentioned as TPCs. Standardize and make it uniform throughput the manuscript.
14. In the conclusions, I noticed that some of the findings and discussions had been repeated. What is the role of identified phytoconstituents in anti-inflammatory action? The author should increase the work's novelty (in the conclusion section). The findings/insights should be used to support this section. Finally, this should give a clear understanding of the study. Future viewpoints need to be covered in the conclusion. The significance of the research should be emphasised by the author.
Author Response
As requested, we replaced the title of the article, as suggested by reviewer 01.
Similarly, we conducted a rigorous evaluation of the abstract and added important details of the research as requested.
In addition, the excessively long lines were reduced, as instructed for lines 22-26 by the reviewer. As for line 26, we have complied with the request, highlighting the existing gaps, which motivated research. As for lines 34-37, the list of compounds presented was reduced and future directions for research were added, as requested by Reviewer 01.
We also made changes in the introduction, following the Reading of the bibliographies suggested by the reviewer who better directed us in relation to the disadvantages of NSAIDs, and we also improved the discussions on the advantages of using the zebrafish model, being of fundamental importance for the work.
Finally, all requests, such as standardization of abbreviations throughout the text, diagramming of the bar of graphs, discussion, conclusion and reference were changed as requested by the reviewers. Still being discussed Lipinkski’s Rule of Five, as requested by the reviewer.
Above all, the manuscript was subjected to extensive proofreading in English by a fluent speaker of the language, with the purpose of correcting the severe compromise of the text indicated above, such as grammatical errors, typos and syntax errors in the manuscript, therefore ensuring the quality and fluidity of the article.
In summary, the suggestions were fully taken on board by the authors.
Finally, we thank you for the indications and we are available for corrections or doubts about any other points of the work submitted to the Journal.
Respectfully,
The Authours.
Reviewer 2 Report
1. The English need improvement since there are some grammatical and syntax errors in the manuscript. For example,
· in line number 106, the word “study” may be as “the study”;
· in line number 162, “control” as “the control”;
· in line number 162, “few” as “a few”;
· in line number 172, “Only” as “The only”;
· in line number 173, “there were” as “there was”;
· in line number 200, “an inhibition” as “inhibition”;
· in line number 269, “tubular” as “the tubular”;
· in line number 286, “similar with” as “similar to”;
· in line number 313, “specific” as “a specific”;
· in line number 332, “a greater” as “greater”;
· in line number 333, “with free” as “with the free”;
· in line number 347, “free” as “the free”;
· in line number 349, “similar in” as “similar to”;
· in line number 350, “higher” as “a higher”;
· in line number 382, “similar with” as “similar to”;
· in line number 472, “of renal” as “of the renal”;
· in line number 481, “excess of” as “excess”;
· in line number 499, “a cellular” as “cellular”;
· in line number 570, “similarity” as “the similarity”;
· in line number 595, “temperature” as “the temperature”;
· in line number 640, “, then” as “, and then”;
· in line number 653, “absence” as “the absence”;
· in line number 735, “Student-Newman-Keuls” as “the Student-Newman-Keuls”;
· in line number 770, “confirmed” as “was confirmed”;
· in line number 775, “high” as “a high”.
The grammar mistakes which are not mentioned here are also to be checked and corrected properly.
2. There are some typing mistakes as well, and authors are advised to carefully proof-read the text. For example,
· in line number 43, the words “off” may be as “of”;
· in line number 44, “the the interactions” as “the interactions”;
· in line number 74, “researches” as “research”;
· in line number 77, “biflavonoids” as “bioflavonoids”;
· in line number 153, “table” as “Table”;
· in line number 215, “intraperineal” as “intraperitoneal”;
· in line number 327, “best evaluated” as “best-evaluated”;
· in line number 355, “hepatoprotector” as “hepatoprotective”;
· in line number 720, “indicates” as “indicate”;
· in line number 755, “pro-teins” as “proteins”.
The typos not mentioned here are also to be checked and corrected properly.
3. Check the abbreviations throughout the manuscript and introduce the abbreviation when the full word appears the first time in the text and then use only the abbreviation (For example, GC-MS, total ion chromatogram (TIC), DMSO, etc.,). And it should be in both abstract as well as in the remaining part of the manuscript. Make a word abbreviated in the article that is repeated at least three times in the text, not all words need to be abbreviated.
4. The full form of the species should be given when the first time appears in both the abstract and in the remaining part of the manuscript and it should be followed by only the first letter of the genus (e.g., Trattinnickia rhoifolia when the first time appears and followed by T. rhoifolia).
5. In the results, figure number 8 should be represented some other forms, since the values is not properly visible to differentiate in 8a and 8c for better understanding.
6. In the result should be only the findings of the present work and not with others. The same may be given in the discussion section. It should be the findings of the present study only and not with the supportive evidence or others. The same may be shifted to either introduction or materials and methods or discussion of the manuscript.
7. The references cited in the text should be as per the journal instruction. For example, “Souza et al. (2016) [17]” may be replaced as “Souza et al. [17]”
8. The statement of Lipinkski’s rule of five may be added, since traditionally, therapeutics have been small molecules that fall within the Lipinski's rule of five.
9. In conclusion, the authors may be included the limitation of the present findings and future direction for a better understanding of the manuscript.
10. The reference should be cited properly, for example, i) few reference full name of the journals given and for others only short form has been given ii) the journal name should be italic, iii) for few reference DOI number is given, but not for other references iv) and all the references to be corrected as per the journal format.
Author Response

(The authors gave the same response as above.)

Round 2
Reviewer 1 Report
In response to my comments, the authors have revised the manuscript. However, I was unable to identify the response for every comment. When responding, the authors should include point-by-point explanations rather than just stating “Finally, all requests, such as standardization of abbreviations throughout the text, diagramming of the bar of graphs, discussion, conclusion and reference were changed as requested by the reviewers”. This form of response is inappropriate.
For example, I was unable to notice any changes in the figures. Figure 5, 7 and 8 – In the figure bar diagram indicated as 100 mg/kg Leaves, 200 mg/kg Leaves and 500 mg/kg Leaves, this is to be changed as 100 mg/kg HELTr, 200 mg/kg HELTr and 500 mg/kg HELTr. This also should be uniform throughout the manuscript.
In addition, only two of the first 24 references listed in the introduction were recently published. The rest are all extremely outdated, with the majority being published between 2000 and 2010. The authors stated that they have referred two of the most recent articles listed in my previous comments 3 and 4, but I was unable to locate those citations in the manuscript. (https://doi.org/10.3390/molecules27030734; https://doi.org/10.3390/molecules27082572)
Apart from that, the manuscript is error-free and might be accepted for publication in the Molecules Journal.
Author Response
All points highlighted by the reviewers were considered in the correction, and the comments on the corrections are in the attached document.

Reviewer 2 Report
1. There are some grammatical, alignment and typographical errors are noted in the manuscript and it should be thoroughly checked and corrected throughout the manuscript.
For example,
· in line number 50, the words “development, but still” may be as “development but still,”;
· in line number 75, “bioactives” as “bioactive”;
· in line number 87, “stand” as “stands”;
· in line number 171, “few” as “a few”;
· in line number 209, “phosphate buffered” as “phosphate-buffered”;
· in line number 267, “in positive” as “in the positive”;
· in line number 267, “i n” as “in”;
· in line number 318, “pose” as “poses”.
2. The authors should do the following in both abstract and the remaining part of the manuscript. Check the abbreviations throughout the manuscript and introduce the abbreviation when the full word appears the first time in the text and then use only the abbreviation. For example, expansion for GC-MS has been given in the abstract but not the rest of the manuscript (in line number 142). This should be checked for all the abbreviations used in the manuscript.
3. The authors have not carried out the following suggestion properly. The full form of the species should be given when the first time appears in both the abstract and in the remaining part of the manuscript and it should be followed by only the first letter of the genus (e.g., Trattinnickia rhoifolia when the first time appears and followed by T. rhoifolia). In line numbers 627 and 791 the full of the plant given again (Trattinnickia rhoifolia). And also for “Danio rerio”.
4. In the results, figure number 8 should be represented some other forms, since the values is not properly visible to differentiate in 8a and 8c for better understanding.
5. The authors have did the following correction, references cited in the text should be as per the journal instruction. For example, “Souza et al. (2016) [17]” may be replaced as “Souza et al. [17]”. But the same has to be checked for other references used in the manuscript. For example, “Borges (2018) [23] and Souza, 2019 [28]” should be replaced as “Borges [23] and Souza, [28]” (in line number 492).
Author Response

(The authors gave the same response as above.)
